# AIM: A network model of attention in auditory cortex

Kenny F. Chou[1,2]*, Kamal Sen[1,2,3,4]*

**1** Department of Biomedical Engineering, Boston University, Boston, Massachusetts, United States of America, **2** Hearing Research Center, Boston University, Boston, Massachusetts, United States of America, **3** Neurophotonics Center, Boston University, Boston, Massachusetts, United States of America, **4** Center for Systems Neuroscience, Boston University, Boston University, Boston, Massachusetts, United States of America

* kfchou@bu.edu (KC); kamalsen@bu.edu (KS)

## Abstract

Attentional modulation of cortical networks is critical for the cognitive flexibility required to process complex scenes. Current theoretical frameworks for attention are based almost exclusively on studies in visual cortex, where attentional effects are typically modest and excitatory. In contrast, attentional effects in auditory cortex can be large and suppressive. A theoretical framework for explaining attentional effects in auditory cortex is lacking, preventing a broader understanding of cortical mechanisms underlying attention. Here, we present a cortical network model of attention in primary auditory cortex (A1). A key mechanism in our network is attentional inhibitory modulation (AIM) of cortical inhibitory neurons. In this mechanism, top-down inhibitory neurons disinhibit bottom-up cortical circuits, a prominent circuit motif observed in sensory cortex. Our results reveal that the same underlying mechanisms in the AIM network can explain diverse attentional effects on both spatial and frequency tuning in A1. We find that a dominant effect of disinhibition on cortical tuning is suppressive, consistent with experimental observations. Functionally, the AIM network may play a key role in solving the cocktail party problem. We demonstrate how attention can guide the AIM network to monitor an acoustic scene, select a specific target, or switch to a different target, providing flexible outputs for solving the cocktail party problem.

## Author summary

Selective attention plays a key role in how we navigate our everyday lives. For example, at a cocktail party, we can attend to friend's speech amidst other speakers, music, and background noise. In stark contrast, hundreds of millions of people with hearing impairment and other disorders find such environments overwhelming and debilitating. Understanding the mechanisms underlying selective attention may lead to breakthroughs in improving the quality of life for those negatively affected. Here, we propose a mechanistic network model of attention in primary auditory cortex based on attentional inhibitory modulation (AIM). In the AIM model, attention targets specific cortical inhibitory neurons, which then modulate local cortical circuits to emphasize a particular feature of

**Data Availability Statement:** All code used for this work are available online at www.github.com/kfchou/AIM_network.

**Funding:** This work was supported by an NSF Award 1835270 (to KS). The funders had no role in

study design, data collection and analysis, decision
to publish, or preparation of the manuscript.

**Competing interests:** The authors have declared
that no competing interests exist.

sounds and suppress competing features. We show that the AIM model can account for
experimental observations across different species and stimulus domains. We also demon-
strate that the same mechanisms can enable listeners to flexibly switch between attending
to specific targets sounds and monitoring the environment in complex acoustic scenes,
such as a cocktail party. The AIM network provides a theoretical framework which can
work in tandem with new experiments to help unravel cortical circuits underlying
attention.

## Introduction

A hallmark of cortical processing is the capacity for generating flexible behaviors in a context-
dependent manner. A striking example of a problem that requires such cognitive flexibility is
the cocktail party problem, where a listener can selectively listen to a speaker amongst other
speakers [1]. Listening in such settings can be highly flexible, depending on the goal of the lis-
tener. For example, a listener can *monitor* the entire auditory scene, *select* a particular target,
or *switch* to another target. Recent theoretical and experimental studies have begun to propose
model networks and cortical mechanisms for producing flexible behaviors [2–8], and top-
down control of cortical circuits via attention is thought to be a critical component.

The influence of attention on cortical processing has been intensively investigated in vision,
resulting in a prominent theoretical framework of attention [9,10]. In contrast, relatively little
is known about attentional mechanisms in auditory cortex. After the early discovery of "atten-
tion units" in the auditory cortex [11], there has recently been renewed interest on attentional
effects in auditory cortex [12–14]. In comparison to the effects of attention in primary visual
cortex, which are relatively small and excitatory [15], attentional effects in primary auditory
cortex (A1) can be much larger and suppressive [13,16,17]. However, a theoretical framework
for cortical mechanisms underlying auditory attention is lacking.

The responses of neurons in A1 can change rapidly when an animal is actively engaged in a
task [8,13,16,17]. For example, cortical neurons with broad spatial tuning curves can sharpen
tuning during attentive behavior [16]; whereas the spectral temporal receptive fields (STRFs)
of cortical neurons with narrow frequency tuning can display the emergence of entirely new
excitatory regions [17] or suppressive effects [13]. Cortical network mechanisms underlying
such diverse attentional changes in tuning remain poorly understood. Changes in cortical tun-
ing can also be driven by competing auditory stimuli in cocktail-party settings, even when an
animal is anesthetized [18,19], suggesting the involvement of both bottom-up and top-down
mechanisms [1,12,20,21].

There is a growing literature on computational models of auditory attention in the context
of auditory scene analysis, as discussed in a comprehensive review [22]. These models can be
grouped into bottom-up or top-down models. Bottom-up models have employed time-fre-
quency representations of sound as an "auditory image" to compute salience maps using static
or temporally evolving features of the image; and predictive coding theory to account for
behavioral results in humans and animals processing auditory scenes [23,24]. Top-down mod-
els have formulated neural processing as spectral-temporal "filters" which extract features
from the auditory image. In these models, attention adjusts the filter characteristics to optimize
the detection and discrimination of targets to explain changes in receptive field properties in
behaving animals [25]. Subsequent models have extended the feature analysis framework to
propose computational principles, e.g., temporal coherence for linking multiple features across
time ("streaming") [26], and incorporated task structure to demonstrate that changes in

receptive field properties during behavior can be specific to task demands [27] as observed experimentally [28]. These previous computational models are all formulated in terms of statistical, signal processing or optimization principles. Thus, a key gap in current state-of-the-art models remains in formulating mechanistic models of *how* cortical computations are implemented by underlying cortical circuits. Indeed, as the review points out, "The field is particularly challenged by the lack of theories that integrate our knowledge of cortical circuitry in the auditory pathway with adaptive and cognitive processes that shape behavior and perception of complex acoustic scenes" [22]. So far, circuit-level models of cortical processing underlying the cocktail party problem have largely focused on bottom-up mechanisms [29,30]. Specific cortical circuit mechanisms underlying top-down attentional changes in cortical responses, and their functional role in solving the cocktail party problem, remain unclear [22].

Here we propose a network model to explain how experimentally observed cortical response properties in A1 could arise from underlying network mechanisms, via the interplay between bottom-up and top-down processes. Central to our network model is attentional inhibitory modulation (AIM), i.e., attention-driven modulation of distinct populations of cortical inhibitory neurons. Specifically, this mechanism relies on disinhibition of bottom-up cortical circuits, mediated via top-down inhibitory neurons, a prominent motif observed in cortex [5–8,31]. We first use the AIM network to model attentional changes in spatial and spectral tuning in auditory cortex [16,17], and then illustrate its potential functional role in solving the cocktail party problem.

## Results

### The AIM network

We began by focusing on spatial processing of multiple sound sources in auditory cortex, extending previous models of bottom-up processing (Fig 1A and 1B). The bottom-up network is made up of integrate-and-fire neurons and implements two key operations–integration and competition. Integration is mediated by broad convergence across spatial channels on the cortical neuron (C, Fig 1A), whereas competition is mediated by inhibition across spatial channels via I neurons (Fig 1B). These bottom-up mechanisms explain two key features observed experimentally in anesthetized or passive animals: broad spatial tuning of cortical neurons to single sounds, and sharpening of spatial tuning in the presence of multiple competing sounds [18,19,29,30].

We extended the bottom-up network to model the effects of attention in the AIM network (Fig 1C). Previous studies have shown that bottom-up cortical representations can be modulated in the attentive state by distinct sub-types of inhibitory neurons. To model such attentional inhibitory modulation, we introduced an additional layer of inhibitory neurons (I2, Fig 1C). I2 neurons can control the spatial and spectral tuning of the cortical neuron in different attentive states, by modulating the activity of E and I neurons in the bottom-up network.

### Attentional changes in spatial tuning

A previous experimental study in cat A1 demonstrated that the spatial tuning of cortical neurons sharpen during attentive behavior [16]. Specifically, the authors observed that A1 neurons exhibited broad spatial tuning when the animal was idle but sharpened their spatial tuning when the animal performed a spatial localization task, as demonstrated by the changes in the azimuth-dependent peristimulus time histograms (PSTHs) (Fig 2, 3rd column). We modeled this attention-induced sharpening effect using the AIM network.

In this simulation, the AIM network consisted of an array of spatial channels tuned to locations between -90˚ and 90˚ azimuth. We then probed the network with broadband noise from

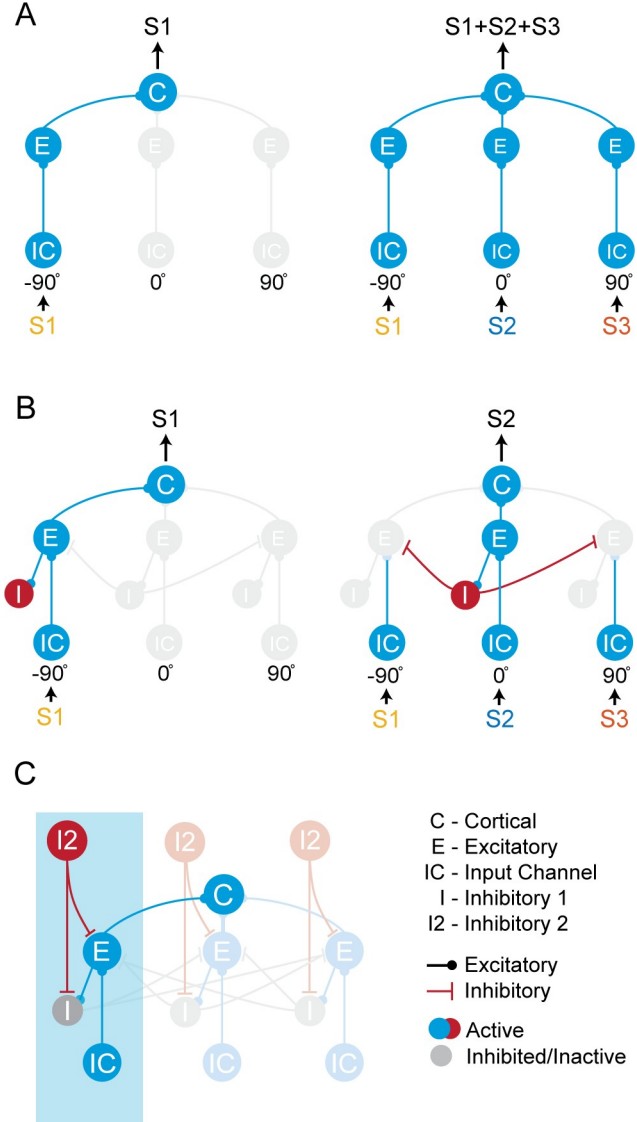

**Fig 1. Sub-networks within the AIM network.** (A) A convergence network. S1, S2, and S3 denote distinct audio stimuli 1, 2, and 3, which are placed virtually at -90˚, 0˚, and 90˚ azimuth, respectively. (B) A passive switching network realized with the addition of I neurons. (C) The AIM network, realized with the addition of I2 neurons, which modulates the I neurons. One spatial channel is highlighted in blue. Neurons in the same spatial channel process the stimuli from a specific spatial location. Response of the cortical neuron, C, represents the output of the network.

various spatial locations and analyzed the C neuron's response as a function of space. Based on the azimuth-dependent PSTHs, we found that when I2 neurons in all spatial channels were on, the cortical neuron in the AIM network exhibited broad tuning—similar to the idle condition in the experiment (Fig 2A). The release of acetylcholine (ACh) during behavioral task performance can suppress intracortical excitatory connections and strengthen thalamocortical connections [32–34]. When we simulated these effects in the AIM network, the spatial tuning of the cortical neuron sharpened, resembling the tuning in the behaving condition in the Lee and Middlebrooks study (Fig 2B). In that study, however, animals were not required to attend to a specific location during the task. We simulated selective attention to a specific location by

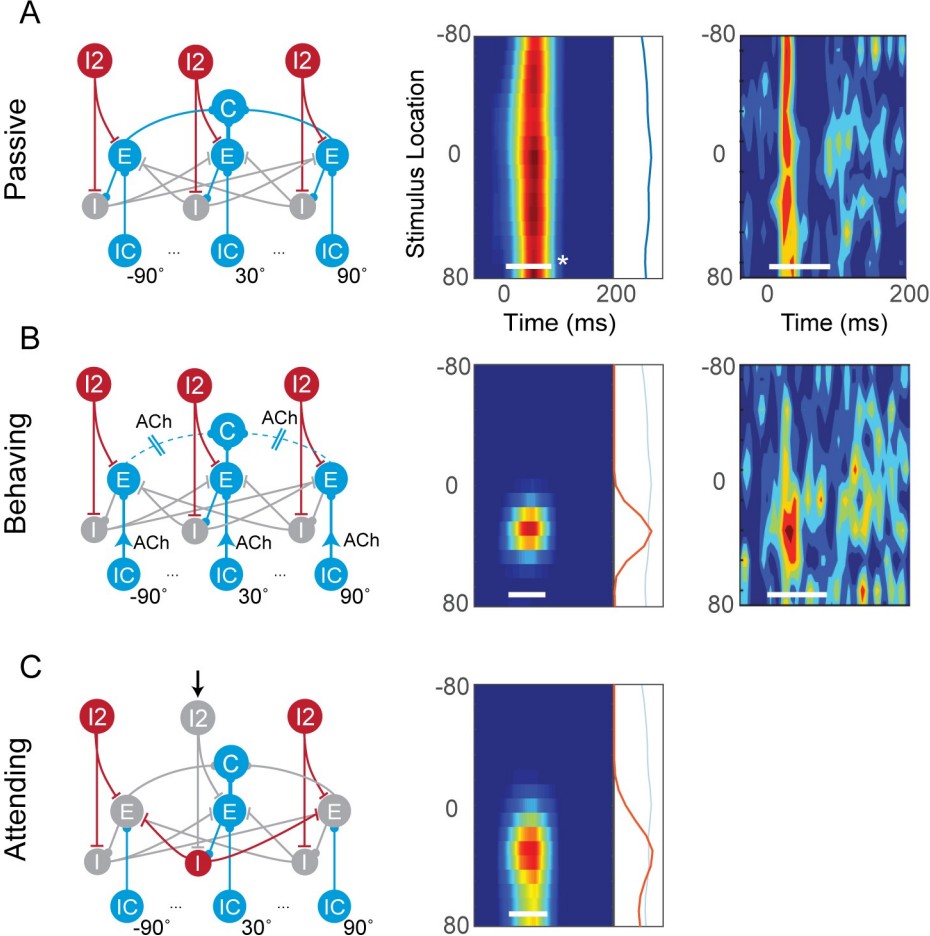

**Fig 2. Attentional sharpening of spatial tuning.** First column shows the spatial AIM network in (A) passive condition; (B) Behaving condition, where the network is modulated with the cholinergic system; (C) Attending condition, where selective attention is simulated by controlling the state of specific I2 neurons. Second column shows the spatial tuning, i.e., PSTH expanded vertically to show the spatial dimension, calculated from the network output. Firing rate is shown (red is higher) as a function of stimulus location over time, in response to a broad band noise. Third column shows experimentally recorded spatial tuning in the cat A1. *White bars indicate the duration of the noise stimulus (80ms). Figures from Lee and Middlebrooks reproduced here with the permission of Nature Neuroscience.

inactivating an I2 neuron in a specific channel, e.g., 30˚ (Fig 2C, left). We found that, in this case, the spatial tuning of the cortical neuron also sharpened (Fig 2C, right). In the AIM network, this effect occurs because of two key mechanisms: the disinhibition of the attended channel by the I2 neuron, which then drives powerful inhibition of competing channels by the I neuron. Thus, selective attention activates focal disinhibition at the attended location and suppression at other locations in the network.

## Attentional changes in spectral tuning

Rapid changes in receptive fields during task performance, thought to arise from attentional mechanisms, have also been observed in the frequency domain as in the experiments by Fritz et al., (2003) and Atiani et al., (2009). Here, we show that the attentional mechanisms in the AIM network can also account for these experimental observations.

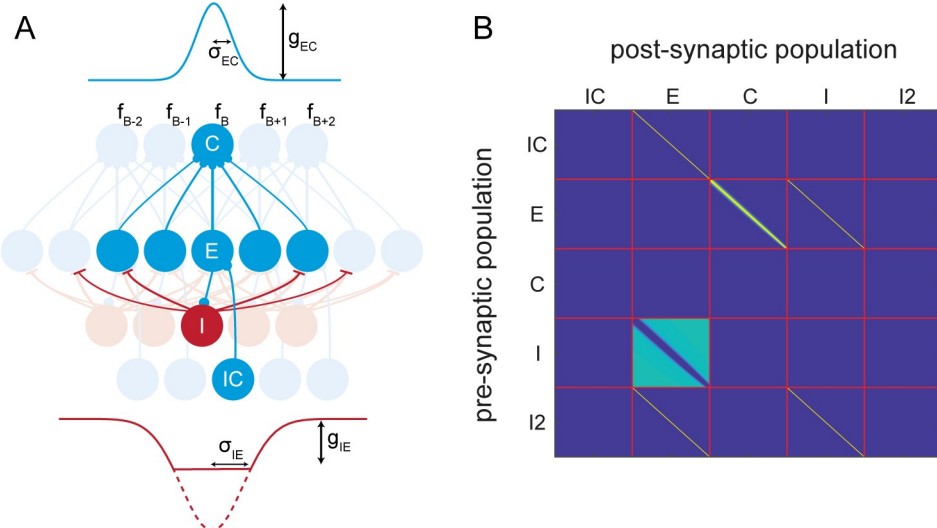

**Fig 3. AIM Network Diagram for the spectral network.** A) convergence of connectivity across neuron types. E neurons converge to C neurons locally, centered around a neuron at the best frequency, $f_B$. The connectivity strength decays across adjacent channels, and is modeled by a Gaussian function. The convergence width is determined by $\sigma_{EC}$, and the connectivity strength is determined by $g_{EC}$ (see Methods). The spread of inhibitory connection from I neurons to E neurons is modeled by a thresholded, inverted Gaussian function. The connectivity within $2\sigma_{IE}$ of this Gaussian function is zero. The inhibition gradually rises to a value determined by $g_{IE}$. B) The network connectivity matrix which describes all connections within the network for these simulations.

We first constrained the AIM network to incorporate experimentally observed features of network connectivity in the frequency domain (Fig 3). Unlike the broad connectivity in the spatial domain (Figs 1 and 2), connectivity across frequency channels is localized to nearby frequencies, reflecting the tonotopic organization of the auditory cortex.

We then simulated the spectral tuning of neurons in the network in the passive vs. attentive states. Here, we probed the network with pure-tone stimuli in the frequency ranges shown in Fig 4. In the attentive state, the network attended to a specific target frequency, distinct from the best frequency of the neuron, as in the experiments by Atiani et al. [35]. Atiani et al. showed the neuron's response as spectrotemporal receptive fields (STRFs). Here, we used frequency-dependent PSTHs as an approximation to STRFs in order to compare our results.

In the attentive state, the I2 neuron in the attended channel (*i.e.*, the target frequency channel, $f_T$, Fig 4) is suppressed during attention, disinhibiting the E and I neurons in that channel (Fig 4A). In this case, we found that when the target frequency was close to the best frequency of the neuron, attention produced an increase as well as a sharpening of the response near the best frequency, as observed experimentally (Fig 4B). In the AIM network, this occurs because when the target frequency is close to the best frequency, excitation from the disinhibited E neuron in the target channel increases the peak response of the cortical neuron, and inhibition via the I neuron in the target channel sharpens the shape of response. In contrast, when the target frequency is far from the best frequency of the neuron, the effect due to inhibition dominates, producing a net suppressive effect on the response, which is also observed experimentally (Fig 4C). Thus, the AIM network qualitatively explains salient features of the experimental observations by Atiani et al.

In addition to sharpening, strengthening, and weakening of STRF hotspots, the experimental results of Fritz et al. showed that a secondary hotspot in the STRF may arise when the animal is in the attentive state. (Fig 5).

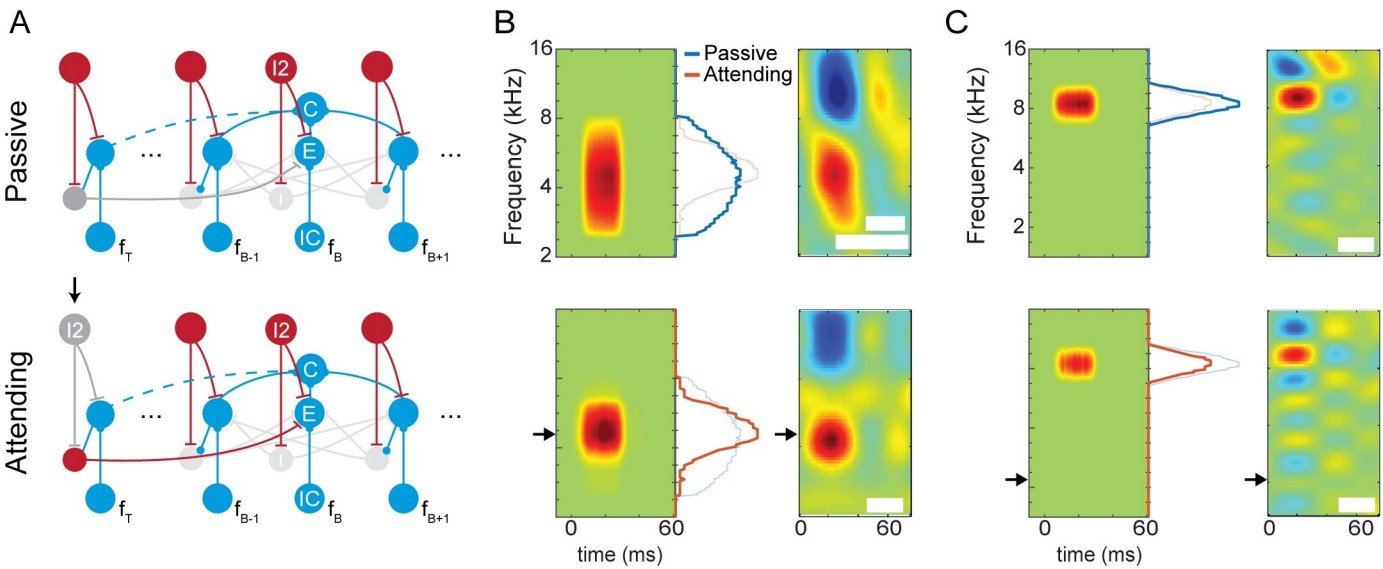

**Fig 4. Simulating attentional effects in Atiani et al.** A) The AIM network in passive and attending state. Black arrow marks the activation of attention on the target frequency channel. The target frequency $f_T$ represents the attended frequency, while $f_B$ represents the neuron's best frequency. Dashed blue line indicates that connections to frequency channels far from $f_B$ are weaker than connections to frequency channels close to $f_B$. B) The results of the AIM network (left panels) versus the results described in Atiani et al. (right panels), when $f_T$ (black arrow) is near $f_B$. Marginals show the total spiking activity across the duration of the simulation. C) The results of the AIM network versus the results shown by Atiani et al., when $f_T$ is far from $f_B$. Figures from Atiani et al. are reproduced here with the permission of Neuron. White patches hide irrelevant text.

We hypothesized that this is a result of the strengthening of an intracortical connection between the target frequency and the best frequency in the attentive state (see *Discussion* for possible mechanisms). To test this hypothesis, we added an additional E-E connection between the $f_T$ and $f_B$, and found that in the passive state, the neuron responded to frequencies near its best frequency, showing a single hotspot. On the other hand, in the attentive state, the same neuron also responded to the target frequency, as seen by the emergence of a new excitatory region at the target frequency in Fig 5B. This effect occurs due to the strengthening of synaptic connection between the target frequency and the best frequency in the attentive state. There is also a suppressive effect on the response to the best frequency, as seen in the slight reduction in amplitude of the tuning curve at best frequency. This effect occurs due to inhibition driven by the I neuron in the target frequency channel. Both of these effects were observed in experimentally measured receptive fields [17]. Thus, the AIM network qualitatively explained salient features of experimentally observed changes by Fritz et al.

## Functional implications

We hypothesized that the attentional mechanisms in the AIM network play an important functional role in processing complex auditory scenes. Two highly effective mechanisms for sound segregation are spatial hearing and frequency selectivity. We first considered functional implications in the spatial domain.

When entering a cocktail party, one might want to monitor the entire scene, focus one's attention on a conversation partner, or switch attention between conversation partners. How does top-down attentional control modulate bottom-up mechanisms to enable this flexible behavior? To illustrate the behavior of the AIM network in these different modes, we simulated several scenarios. In these simulations, we used three spatial channels corresponding 0˚, 90˚,

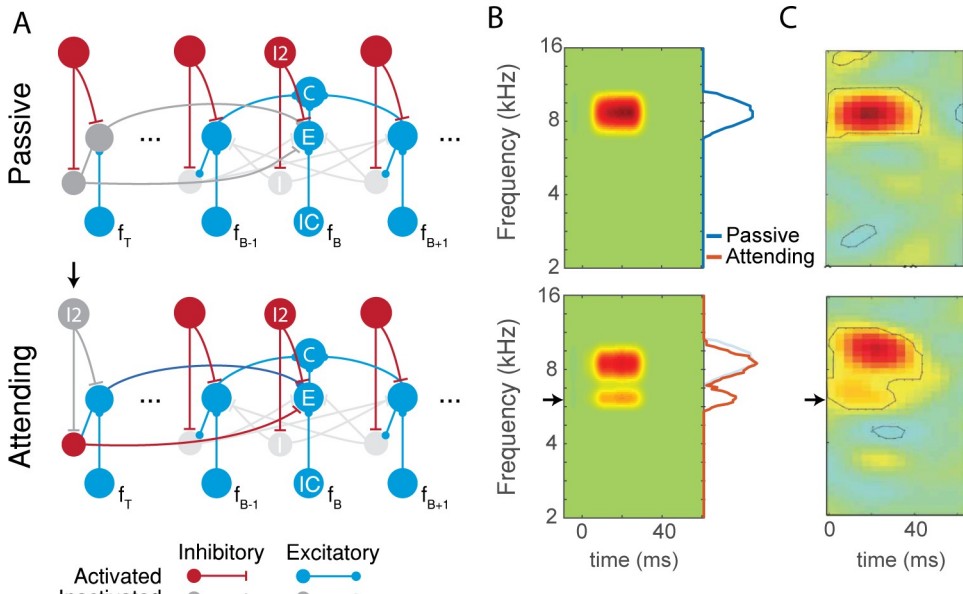

**Fig 5.** Simulating the effects in Fritz et al., A) The proposed mechanism underlying the changes in A1 behavior. Frequency channels adjacent to the best frequency $f_B$ as well as the target frequency $f_T$ are shown. Black arrow indicates the target of selective attention. The dark blue connection highlights the additional intracortical connection unique to this simulation. B) Simulated A1 neuron STRFs in the two states of attention. The model qualitatively reflects the changes observed in physiological recordings. C) Physiological recording of a STRF of an A1 neuron when the animal is passively listening (top) and when the animal is attending to a target tone, marked by the black arrow (bottom). Figures from Fritz et al. are reproduced here with the permission of Nature Neuroscience.

and -90˚ azimuths, presenting the network with different tokens of speech stimuli at these locations, either sequentially or simultaneously (see Methods).

To demonstrate passive listening (the "monitor" mode), we set all I2 neurons active, thereby silencing all I neurons (Fig 6A). In this case, when the speech tokens were presented sequentially to the network, the network output resembles each individual speaker (Fig 6A). When the speech tokens were presented simultaneously, the network output resembles their mixture. Thus, in this mode, the network broadly monitors the acoustic scene across different spatial locations.

To simulate selective attention to a particular speech token, we first inactivated the I2 neuron in the 0˚ spatial channel, thereby activating lateral inhibition via the disinhibition of I neurons in that channel (Fig 6B). In this case, the network output resembled the 0˚ speaker output, regardless of whether the speakers were presented sequentially or simultaneously. Finally, to simulate switching attention to a different location (90˚), we turned off the I2 neuron at 90˚, activating lateral inhibition via the disinhibition of I neuron at that location (Fig 6C). In this case, the network output was more similar to the output for the speaker at 90˚ (Fig 6D).

Finally, we demonstrate the effect of speaker separation on network performance. First, we expanded the network to have seven spatial channels tuned from 0˚ to 90˚ azimuth in 15˚ increments. We then kept the first speaker at 0˚ while varying the location of the second speaker, and presented both speakers to the network simultaneously. We found that the network output became more representative of the attended targets (either speaker 1 or speaker 2) as the two speakers separate in space, demonstrating the effect of spatial release from masking [36] (Fig 6E). In summary, when an I2 neuron in a specific spatial channel is inactivated, it disinhibits the I neuron at that location, causing the network to selectively attend to that spatial

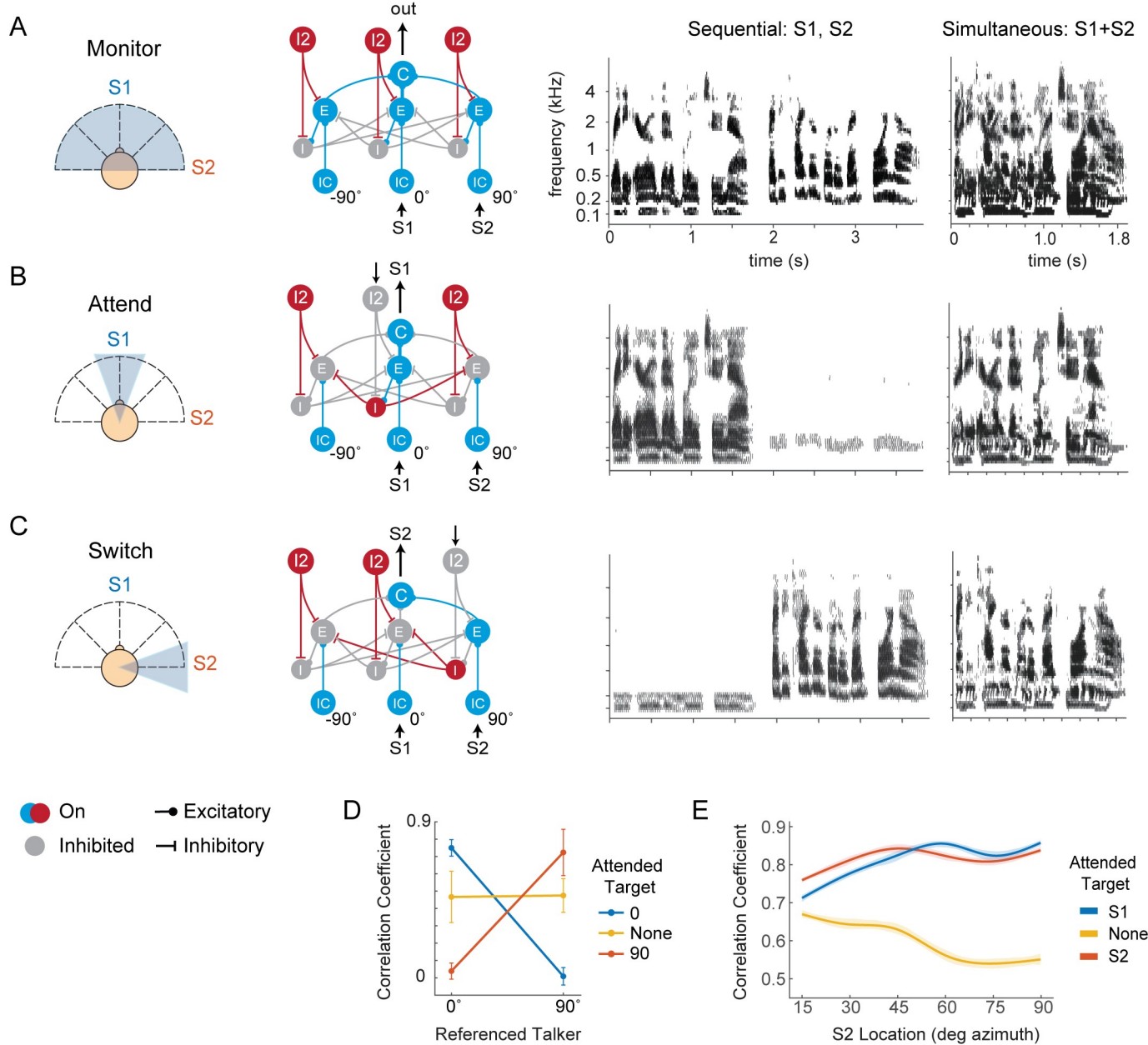

**Fig 6. Functional implications of the AIM network: Spatial tuning of the network is dictated by the state of TD neurons.** (A) The network monitors the entire azimuthal plane when all TD neurons are active. (B) The network attends to a specific direction if the corresponding TD neuron is off. (C) The network attends to a different location if a different TD neuron becomes inactive. Column 3 shows the result of simulations when speakers are presented sequentially to the network, in spike rasters. Column 4 shows the result of simulations when speakers are presented simultaneously to the network. (D) The AIM network can recover an attended target within a speech mixture, as quantified by the cross-correlation measures between the simultaneous simulation network output and single speaker spike rasters. Error bars show standard deviation (n = 20). X-axis is the reference speaker, and each line color denotes the attended location. (E) Spatial separation of two talkers (S1, S2) vs. network performance, as quantified by correlation between the network output to the attended target. The encoding becomes more representative of the attended speaker as the separation between the two speakers increases. In the "not attending" case, S1 is used as the reference for correlation calculation. Shaded area represent 95% confidence interval, n = 20.

location. Additionally, the ability of the network to attend to a target depends on the spatial separation between the target and its maskers.

We next investigated the functional role of the AIM network in the frequency domain using the same network shown in Fig 4. In this simulation, two competing speech tokens, a

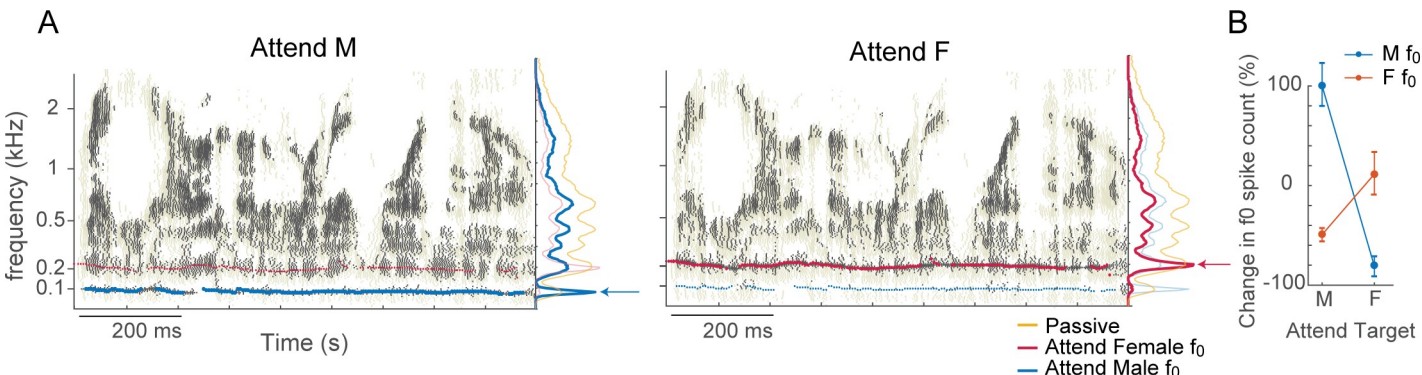

**Fig 7. Functional role of the AIM network in the frequency domain.** (A) AIM network outputs (raster plots) when attending to the male talker's $f_0$ (left) or female talker's $f_0$ (right). Yellow rasters in the background shows the network output in the passive condition. Blue and red lines mark the estimate $f_0$ for each speaker. Blue, red, and yellow lines in the marginal show the total spike counts per frequency channel for attend male, attend female and passive conditions. (B). Change in spike count (%) in the male or female $f_0$ channel when attending to the male or female target, compared to the passive condition. Error bars show standard deviation (n = 20).

male and female speaker, originate from the same spatial location. In this case, differences in spatial cues cannot be exploited for segregation, but differences in spectral features of the two speakers, e.g., the fundamental frequency (F0), are available. We simulated the network in three conditions: passive, attending to male F0, or attending to female F0. Attention was simulated (as in Fig 2) by inactivating the I2 neuron which disinhibits the I neuron at the attended frequency channel. In the attending modes, network activity showed a sharp peak around the F0 of the attended speaker, and a suppression of activity around the F0 of the competing speaker (Fig 7). Similar to the earlier results shown in Fig 3, the attended F0 received an increase in spiking activity in that region while spiking activity in frequencies far from the F0 are suppressed (Fig 7A marginals and Fig 7B).

## Discussion

The capacity for generating flexible behaviors in a context-dependent manner is central to many complex cognitive tasks. How cortical circuits achieve such flexible computations is a central area of investigation in both theoretical and experimental neuroscience. Recent theoretical studies have begun to propose model networks capable of producing flexible behaviors, e.g., gating mechanisms for flexible routing of information [2–4], and experimental studies have begun to reveal cortical mechanisms underlying flexible gating of information and attentional control [4,6,37]. However, such models are lacking for auditory cortex. In this study, we propose the AIM network, which describes a mechanism of interaction between top-down and bottom-up processes in auditory cortex that may underlie the attention-driven changes in cognitive behavior.

### Flexible cortical processing

The rapid flexibility of the AIM network is generated by top-down inputs, which control the state of the network by dictating the on/off state of specific I2 neurons. The top-down inhibition of I2 neurons disinhibits the I neuron of the attended channel, which then suppresses competing channels via top-down lateral inhibition, resulting in focused attention. On the other hand, when I2 neurons in all channels are active, the network integrates information from all input channels. In the spatial case, this produces broad spatial tuning and allows the network to monitor the entire scene. The ability to switch between these behaviors is important from a functional standpoint. A network that is always selective for a single channel may

fail to detect important events in the scene at other locations. The different states of the I2 layer (e.g., all on vs. one off), allows the opportunity for *exploration* (by detecting events across a broad range of locations) as well as *exploitation* (by selectively listening to a particular channel) in a dynamic manner.

Switching between exploration and exploitation is especially interesting in the context of findings that the "spotlight of attention" fluctuates in a rhythmic manner [38]. In our model, such fluctuations in the strength of the top-down input would cause the network to alternate between periods favoring broad detection of sounds across the entire acoustic scene and fine discrimination at a single spatial location [39]. Such periods may allow salient sounds in the background to capture the spotlight of attention, as demonstrated in a recent study in humans [40]. Moreover, changes in the location of the top-down input would promote switching attention to a different location [41]. These various behaviors may play important roles in how animals navigate complex environments.

## Alternative/additional mechanisms

The AIM network describes the mechanistic interaction between top-down and bottom-up processes, and even though it can explain various experimental observations, alternative/additional mechanisms may be involved in some aspects of the experiments modeled here.

**Neuromodulation.** The neuromodulatory systems can exert powerful control over the *global* state of cortical networks, e.g., asleep, quiet arousal, and active attention [42]. Two key modulatory projections to cortex involve norepinephrine (NE) and acetylcholine (ACh), which have been implicated in arousal and attention, respectively. In our network, we assumed that when the network is in the "monitor" mode, the I2 layer is on. Such a global state may correspond to arousal and be NE-dependent. Top-down suppression of I2 neurons in our model could correspond to an attentive state and be ACh-dependent.

Indeed, we showed that global cholinergic effects on cortical circuits, i.e., suppression of intracortical connections and enhancement of thalamocortical connections, could also produce a sharpening in spatial tuning (Fig 2). In the study by Lee and Middlebrooks [16], animals were not required to attend to a fixed location when performing the localization task. Thus, sharpening of attention via the activation of global cholinergic mechanisms may be more consistent with that experimental design.

**Top-down inputs from frontal areas.** Although cholinergic mechanisms are clearly important in attentional states, such mechanisms are thought to operate on slow timescales and can be long lasting [42], whereas switching between exploration and focused attention requires rapid, reversible changes in cortical outputs, potentially on sub-second timescales. Recent studies with EEG and fMRI in humans have suggested top-down activation in the frontal areas modulates processing in auditory cortex on the time scale of hundreds of milliseconds [43,44]. Evidence from studies in mice also support the idea that the frontoparietal network can modulate processing in the primary sensory cortices during selective attention [5,45–47]. Together, these studies suggest that the top-down signals responsible for modulating the AIM network may originate from the frontal areas.

**Synapse-specific gating.** Our model predicts that experiments where animals are required to selectively attend to a specific location should also produce a sharpening of spatial tuning in A1. For the experiments by Fritz et al. [17], we found that strengthening of the intra-cortical synaptic connection between the target frequency and the best frequency could explain the emergence of new excitatory regions at the target frequency. A possible mechanism for transient, reversible strengthening of intracortical synapses is synapse-specific gating [3,4], which may then promote long-term strengthening via classic Hebbian plasticity.

## Cortical inhibitory neurons and function

Inhibitory neurons play key roles in the AIM network. There are several types of inhibitory neurons in cortex [48]. The majority of inhibitory neurons can be placed into three categories: those that express parvalbumin (PV), somatostatin (SOM), or vasointestinal peptide (VIP). It is worth noting that most currently available information on specific classes of interneurons come from studies in rodents in a variety of cortical areas, whereas key experimental observations modeled in this study were obtained in other species. Thus, it is difficult to directly map the functional groups of neurons, e.g., I2 and I, in the model to identified interneuron types e.g., PV, SOM and VIP neurons. Nevertheless, we suggest some hypotheses on a possible correspondence based on recent experiments in rodents, to motivate future experimental work.

**VIP neurons.**   The top-down input in the model could correspond to inputs from VIP neurons. VIP mediated inhibition is engaged under specific behavioral conditions, including attention [5,7]. It has been proposed that VIP cells "open holes in the blanket of inhibition" [49], generating the "spotlight of attention" [5]. Our results are consistent with this intuition, with the top-down input being critical for selecting a particular target and switching to a different target.

VIP input is often thought to favor excitation, due to the disinhibition of excitatory neurons [50]. In the model, top-down inhibition of an I2 neuron in a specific channel activates powerful inhibition via I neurons that suppress competing channels, leading to the selection of the target. Thus, the model also explains powerful suppressive effects of selective attention, which have been observed in auditory cortex [13]. The model predicts that silencing top-down inputs to a specific channel, via optogenetics or other methods, should block the effects of selective attention.

**SOM and PV neurons.**   VIP neurons are known to inhibit SOM neurons, which in turn inhibit excitatory neurons [31]. This motif suggests that the I2 neurons in our model may correspond to SOM neurons, specifically Martinotti cells, which are strongly targeted by VIP neurons [51].

The I neurons mediate powerful and sustained inhibition of competing channels in the model. A key distinction between this type of inhibition and "classical" lateral inhibition observed at multiple stages of sensory processing starting at the periphery is noteworthy. Classical lateral inhibition is activated by bottom-up stimulus-driven mechanisms, whereas the inhibition in our model is driven by top-down attentional mechanisms. To distinguish these two cases, we refer to the inhibition mediated by I neurons in the model as "top-down lateral inhibition". This distinction is conceptually and functionally important, because unlike classical lateral inhibition, which is recruited automatically by the stimulus, top-down lateral inhibition can be recruited volitionally. Such top-down lateral inhibition can be activated by direct disinhibition of I neurons, disinhibition of E neurons which drives feedback lateral inhibition via I neurons (S1 Fig), or a combination of the two.

In principle, top-down lateral inhibition could be mediated by any interneuron type. Although PV neurons are a possible candidate, the long-lasting inhibition required to suppress competing channels in our model should be distinguished from the fast and transient dynamics of inhibition typically associated with PV neurons [48]. SOM neurons can also mediate feedback lateral inhibition to generate a "winner-take-all" circuit and suppress competing channels [52], or modulate bottom-up inputs in specific layers [51,53]. Developing behavioral paradigms for investigating attention in rodents combined with optogenetic manipulations, and/or developing methods for selectively manipulating different interneuron types in other species, are promising future directions for identifying specific cell-types involved in mediating top-down lateral inhibition.

## Space vs. frequency

We related the effects of attention in the AIM network to key experimental observations on changes in cortical spatial and frequency tuning in animals engaged in a behavioral task vs. passive animals [16,17]. Similar changes have also been reported in the primary auditory cortex of humans [54]. Assuming that attentional mechanisms are a key factor in driving such changes [12,16,55], the effects of attention appear very different in the spatial and frequency domains. In the spatial domain, broad tuning sharpens during task performance, whereas in the frequency domain, narrow tuning can be enhanced or suppressed depending on the target frequency and other parameters such as SNR. Our results suggest these apparent differences in the spatial vs. the frequency domain may share similar underlying attentional mechanisms.

A key difference between the spatial and frequency domains in our model is the convergence from the E neurons to the C neuron, which is broad in the spatial domain but narrow in the frequency domain. Previous studies in the auditory cortex have found a tonotopic organization, but no topographic organization for spatial tuning [56]. Therefore, local synaptic connections in a patch of cortex may result in convergence from neurons with similar tuning in frequency but a broad range of tuning in space, which is consistent with our model. Thus, our results suggest that the same cortical mechanisms underlying attention can produce diverse effects on stimulus tuning, due to differences in the cortical organization of stimulus features, e.g., space or frequency. For simplicity, we considered spatial and spectral networks separately. Future models should unify these two dimensions.

## Domain-specific considerations

**Spatial domain.**   Here, we did not explicitly model how spatially tuned inputs to the AIM network arise, an aspect that is likely to be species dependent. In the AIM network, spatial tuning is inherited from tuning for acoustic cues in pre-cortical areas [30], perhaps the simplest scenario consistent with experimental observations [57–59]. Additional mechanisms, e.g., forward suppression, may further sharpen or generate spatial tuning in cortex [60,61]. In rodents, spatially tuned responses covering a range of azimuths have been observed in cortical areas [62], and may emerge from excitatory-inhibitory interactions in the underlying network [63]. From a functional standpoint, it is interesting to note that sharp tuning is not necessary for the monitor mode, but only for the selective mode of the AIM network. In some species, the sharpness of tuning may emerge in the attentive state based on state-dependent mechanisms, and/or inputs from other brain areas, e.g., the superior colliculus, which shows a map of auditory space [64,65] and can modulate responses in A1 via the pulvinar [66]. These outstanding issues will require further experimental work, especially in attentive animals, as well as the development of species-specific models.

**Spectral domain.**   The model has several simplifications and limitations that motivate future directions of work. For example, in the frequency network, we used pure tones to characterize the responses of neurons in the network and relate them to experimentally observed STRFs obtained with ripple noise stimuli. Although this approach captured salient attentional effects observed experimentally, future studies should probe non-linear components using complex stimuli, e.g., ripples and natural sounds.

In this study, we focused on excitatory regions of STRFs, modeling three representative changes in the excitatory regions of STRFs observed by Fritz et al and Atiani et al. [35]. However, it is known that the balance between both excitatory and inhibitory subregions can play an important functional role [67]. In preliminary simulations we found that attention could also decrease an inhibitory subregion, as observed by Fritz et al. (S2 Fig). Modeling the effects of attention on inhibitory regions, and complex STRFs with both excitatory and inhibitory

regions, merit further investigation in the future. For example, the diverse effects of training paradigms on inhibitory regions [28], will require modeling the effect of training, reward or punishment on cortical circuits in the model.

We used the AIM network to illustrate how top-down inhibition of I2 neurons can enhance the representation of sounds at an attended location, or an attended feature, e.g., F0 in speech, by suppressing competing sounds at a different location or F0. F0 is likely to be one of many potential features that contribute to speech segregation. However, a similar principle could be applied to more complex features, e.g., enhancing the representation of an attended harmonic, employing harmonic template neurons in the auditory cortex [68]. Such enhancement of an attended feature accompanied by suppression of competing features may contribute to speech segregation in settings where spatial cues are unavailable [69].

**Temporal domain.** Temporal dynamics likely play a large role in auditory processing. For example, the neurons observed by Lee and Middlebrooks are highly sensitive to stimulus onset (Fig 2). Our work focused on the effect of attention in the spatial and spectral domain, and thus we did not include detailed models of temporal dynamics, e.g., adaptation or sensitivity to stimulus onset or offset, or investigate the transient tuning properties during attention switching. Additionally, we did not investigate temporal phenomena that is likely to play an important role in speech segregation, e.g., temporal aspects of F0 or tracking of slow spectrotemporal modulations [70], and auditory "streaming" for linking sound segments over time [71]. We believe that thoroughly investigating these aspects at the cortical circuit level will require modeling rich temporal aspects of neuronal and network dynamics, e.g., adaptation, synaptic facilitation and depression, oscillations, synchrony, and coherence, and is outside the scope of this current study. Future extensions of the AIM network should incorporate these aspects to link mechanisms of neuronal and network dynamics to attentional dynamics.

## A mechanistic model of attention

Previous studies have modeled the effects of attention on auditory cortical receptive fields using mathematical and computational principles such as temporal coherence [22,72]. In contrast, the AIM network is a cortical circuit level model underlying attentional effects. One recent study modeled different STRFs in the attending vs passive state of the ferret A1 with a two-layer spiking network [73]. The focus of that study was to produce detailed fits of STRFs in attending and passive animals. In contrast, the focus of this study was to propose general cortical circuit mechanisms, e.g., top-down disinhibition, underlying the effects of attention on both spatial and spectral tuning. Another previous study modeled global cholinergic mechanisms underlying changes in STRF [74], similar to the effects modeled in Fig 2B. However, that study did not include the selective top-down disinhibitory mechanism, which was unknown at that time and is a key mechanism in the AIM network.

Original models of attention in vision were also developed based on computational principles, e.g., biased competition or normalization [9,10]. At the time, available information on cortical circuits to guide and constrain circuit-based models were limited. Subsequently, cortical circuit-based models of visual attention have been proposed [75]. With the rapidly emerging knowledge of specific cell types and circuitry in auditory cortex, along with the availability of powerful optogenetic tools for cell type-specific perturbations, the AIM network may help guide the design of new experiments to unravel cortical circuits that underlie general attention.

## Methods

Simulations and models were implemented in Matlab (Natick, MA, USA).

## Stimuli

Three sets of auditory stimuli were used, depending on the specific simulation. White Gaussian noise was used as the stimulus in spatial tuning simulations, pure tones with frequencies approximately equal to the center frequencies of the gammatone filterbank (see Subcortical Processing) were used in spectral tuning simulations, and speech tokens from the Coordinated Response Measure (CRM) corpus were used in the functional demonstration simulations [76]. In spatial simulations where stimuli were placed along the azimuth, directionality is imparted on the stimuli by convolving them with the head-related transfer functions (HRTFs) of the Knowles Electronics Mannikin for Acoustic Research (KEMAR) [77,78].

## Subcortical processing

Stimuli for each simulation were first processed and encoded with models of the auditory periphery and midbrain, then presented to the network. The auditory periphery was modeled by a gammatone filterbank, implemented using the Auditory Toolbox [79]. It was used to separate the sentence mixture into 64 narrowband frequencies, with center frequencies ranging from 200 to 8000 Hz, uniformly spaced on the equivalent rectangular bandwidth scale.

We used a previously published model of the midbrain to perform spatial segregation of spatialized stimulus mixtures, as well as to encode the stimuli. If a simulation did not use spatial stimuli as the input, the stimuli were treated as dichotic. For details pertaining to the midbrain model, see Fischer et al. and Chou et al. [30,80]. Briefly, the midbrain model computed binaural features (i.e., interaural timing and level differences) in each time-frequency tile (i.e., narrowband and short time window). Model neurons encoded the stimulus at specific time-frequency tiles if the binaural features of the stimuli matched the "preferred" binaural features of the model neuron, thereby performing spatial segregation. The preferred binaural feature of each model neuron is specific to the frequency and spatial channel each neuron belonged to. There were 64 frequency channels in the midbrain model, corresponding to each channel of the gammatone filter. The number of spatial channels in the midbrain model depended on each specific simulation. The input neuron in a spatial channel is spatially tuned to the azimuth corresponding to that channel, consistent with spatial tuning of acoustic cues observed in subcortical areas [57–59]. The spiking responses of these model neurons were used as the input to the AIM network.

## Attentional inhibitory modulation (AIM) network

The AIM network was implemented using the DynaSim package [81], and its structure is illustrated in Fig 1. For simplification purposes, only one frequency channel and three spatial channels are shown. A "spatial channel" refers to the sub-network of neurons that are responsible for processing inputs from a specific spatial location (blue shading, Fig 1). The number of spatial and frequency channels in the network, and their connectivities, depended on the specific simulation being explored.

Five neural populations were created within the network: excitatory input (IC), excitatory (E), inhibitory (I), output cortical (C), and a second inhibitory (I2) population. IC neurons represent the bottom-up inputs to the network from the subcortical model. I2 neurons represent attentional top-down control. With the exception of the C neurons, a number of neurons were created within each population, corresponding to each of the spatial or frequency channels needed in a simulation. All five neural populations are implemented as leaky integrate-and-fire neurons whose dynamics are defined by The following differential

equation [82]:

$$\frac{dV}{dt} = \frac{(g_{leak}(E_{leak} - V) - i_{syn})}{C}$$

where $V$ is the membrane potential, $i_{syn}$ is the synaptic input current, $C$ is membrane capacitance, $g_{leak}$ is the membrane conductance, and $E_{leak}$ is the equilibrium potential. The spike-and-reset mechanism employed in our model dictates that if $V > V_{thresh}$, then $V \rightarrow V_{reset}$. Here, $V_{thresh}$ is the action potential threshold and $V_{reset}$ is the reset voltage. Values for these parameters are listed in Table 1.

The dynamics of the synaptic input current is defined by a double exponential:

$$i_{syn}(t+1) = i_{syn}(t) + g_{syn}\left[\left(e^{-\frac{t}{\tau_D}} - e^{-\frac{t}{\tau_R}}\right)u(t)\right](netcon)\left(V - E_{syn}\right) + i_{app}(netcon)$$

where $t$ is time since the previous spike, $g_{syn}$ is the synaptic conductance, $\tau_D$ and $\tau_R$ are the decay and rise time constants, respectively, and the difference of exponentials represent the excitatory post-synaptic potential (EPSP) waveform. $u(t)$ is the unit step function to ensure that EPSP is zero before the previous spike has occurred. $E_{syn}$ is the reversal potential, $i_e$ is the externally applied current, $i_{app}$ is the externally applied current, and $netcon$ refers to a binary matrix of network connectivities that define the connections between populations of neurons. Each row in the $netcon$ matrix represents a presynaptic neuron, and each column represents a postsynaptic neuron. Binary entries of $netcon$ represents presence of a synaptic connection between neurons. Inhibitory synapses have the following parameters: $\tau_R = 1ms$, $\tau_D = 10ms$, $E_{syn} = -80mv$. Excitatory synapses have the following parameters: $\tau_R = 0.4ms$, $\tau_D = 2ms$, $E_{syn} = 0mv$. The values for $g_{syn}$ and $i_{app}$ are simulation- and connection-dependent, and are listed in Table 2. The network connections are illustrated in Figs 1 and 3.

The default $g_{syn}$ were chosen such that if I neurons were off, then the inputs would be relayed and combined at the C neuron with a similar firing rate, and if I neurons were on, then E neurons would be completely silenced.

## Simulation-specific model configurations

**Lee & Middlebrooks simulations.** In this spatial tuning simulation, 80 ms of white gaussian noise was placed between -80° to 80° azimuth, in 10° increments. The spatialized stimuli were then processed and encoded with the subcortical model. The midbrain model in this simulation consisted of 19 spatial channels from -90° to 90° azimuth, in 10° increments, and 64 frequency channels. To reduce the computational demand of simulating the AIM network, a new set of spike trains, generated using a Poisson model based on the overall firing rate across

**Table 1. Default parameters of cellular dynamics.**

| Parameter | Value |
| --- | --- |
| $C$ ($nF$) | 1 |
| $g_{leak}$ ($\mu S$) | 0.1 |
| $E_{leak}$ ($mV$) | -70 |
| $V_{thresh}$ ($mV$) | -55 |
| $V_{spike}$ ($mV$) | 50 |
| $V_{reset}$ ($mV$) | -75 |
| $i_{app}$ ($\mu A$) | 0 |
| $Noise$ | 0 |

**Table 2. Simulation-specific parameters.** $g_{syn}$ have units of $\mu S$ and $i_{app}$ have units of $nA$. Parameters of local convergence $g_{IE}$, $g_{EC}$ $\sigma_{IE}$, and $\sigma_{EC}$ are also shown (Fig 3).

| Connection or Neuron | Param | Lee | Fritz | Atiani a | Atiani b | Spatial Function | Freq Function |
|---|---|---|---|---|---|---|---|
| IC→E | $g_{SYN}$ | 2.5 | 4 | 4 | 3 | 2 | 4 |
| E→C | $g_{SYN}$ | 2 | 1.25 | 1.25 | 3 | 2 | 1.5 |
| E→I | $g_{SYN}$ | 2.5 | 3 | 3 | 3 | 2 | 3 |
| E→E (Attend) | $g_{SYN}$ | - | 5 | - | - | - | - |
| E→E (Passive | $g_{SYN}$ | - | 3 | - | - | - | - |
| I2→I | $g_{SYN}$ | 4 | 3 | 3 | 3 | 2.25 | 3 |
| I2→E | $g_{SYN}$ | 2.8 | 4 | 4 | 3 | 1.25 | 3 |
| I→E | $g_{SYN}$ | 3 | 4 | 4 | 4 | 3 | 3.5 |
| I | $i_{app}$ | 3 | - | - | - | 4 | - |
| E | $i_{app}$ | 1 | 3 | 3 | 3 | 0 | 3 |
| I2 | $i_{app}$ | 8 | 8 | 8 | 8 | 3.5 | 8 |
|  |  |  |  |  |  |  |  |
| $g_{EC}$ | - | - | 1 | 1 | 1 | - | 1 |
| $g_{EC}$ | - | - | 1 | 1 | 1 | - | 1 |
| $\sigma_{IE}$ (kHz) | - | - | 0.65 | 1.7 | 2 | - | 0.01 |
| $\sigma_{EC}$ (kHz) | - | - | 0.32 | 1.15 | 0.35 | - | 0.05 |

all frequency channels, were computed for each spatial channel. This operation essentially collapses the neural response over the frequency dimension. Therefore, the AIM network for this simulation consisted of 19 spatial channels and one single frequency channel, where each spatial channel processed the set of spike trains that represent the average activity across all frequencies. Network connectivities between spatial channels are as shown in Fig 2. Spatial tuning curves were then calculated based on the response of the C neuron of the AIM network.

The effects of neuromodulators were simulated by applying a gain on the network connections. During the behavior state, off-target E-C connectivities were applied a gain of 0 to simulate the effects of muscarinic receptors, and off-target IC-E connectivities were applied a gain of 2.5 to simulate the effects of the nicotinic receptors. These gains were chosen to replicate the effects observed experimentally.

**Atiani et al. and Fritz et al. simulations.** Pure tones were presented dichotically to the subcortical model, which consisted of a single spatial channel, corresponding to 0˚ azimuth, and 64 frequency channels. Spike trains were passed directly to the AIM network, which also consisted of a single spatial channel and 64 frequency channels. Network connectivities between spatial channels are as shown in Fig 3. Approximations to spectral temporal receptive fields were calculated based on the response of the best-frequency cortical neuron to each of the pure tone stimulus.

**Calculation of frequency- and azimuth- dependent peristimulus time histograms (PSTHs).** To Approximate STRFs of cortical neurons, we show the responses of the model cortical neuron as functions of time and either frequency or space. In the spatial case, white Gaussian noise were used as stimuli. In the spectral case, pure tones were used as stimuli. Model neuron response for each frequency or azimuth were shown as its firing rate, which was calculated using a 5ms moving window.

**Functional example–spatial listening.** In this example, we demonstrate how the AIM network can be used to isolate a specific talker of interest within a speech mixture. 20 pairs of speech tokens, one male and one female, were randomly chosen from the CRM corpus. The male token was placed at 0˚ and the female token was placed at 90˚ azimuth. For simultaneous

presentation, speech tokens were summed prior to being processed by the subcortical model. The subcortical model used 5 spatial channels, tuned from -90˚ to 90˚ azimuth in 45˚ increments, and 64 frequency channels, and spatially segregates the speech tokens. Its output is relayed directly to the AIM network, which also has 5 spatial channels and 64 frequency channels. In this simulation, network connectivities across spatial channels and parameters are as shown in Fig 2, and each frequency channel operated independently of each other.

To demonstrate the effect of spatial separation on model performance, the male speaker was placed at 0˚ azimuth while the female speaker was placed at locations between 15˚ and 90˚ azimuths in 15˚ increments (Fig 6E). The subcortical model for this simulation used 7 spatial channels, corresponding to locations from 0˚ to 90˚ azimuths in 15˚ increments, and 64 frequency channels. The output is then processed with the AIM network, which has 7 spatial channels corresponding to the same locations in space. The AIM network was set to 1) attend to the male target at 0˚ or 2) attend to the female target at various locations or 3) to be in the monitor mode. The network performance was measured by "similarity" between the network outputs and the attended speakers. In the monitoring mode, male speaker was used as the reference talker. Similarity was quantified by calculating the two-dimensional correlation coefficient between the network output of the specific simulation and the network output of the reference speakers. More specifically, we first calculated the firing rates for each frequency channel, then calculated the two-dimensional correlation coefficient of the firing rates.

**Functional example–monaural listening.** In this example, we demonstrate that the AIM network can also operate in the spectral domain to aid in sound segregation during monaural listening. The same 20 pairs of speech tokens as above were summed and presented dichotically to the subcortical model. Here, both the subcortical model and the AIM network has one spatial channel (0˚ azimuth) and 64 frequency channels. In this simulation, the network connectivities are as shown in Fig 3. The pitch of each speech token was estimated using MATLAB Audio Toolbox's pitch() function. The I-E connectivity parameters were chosen based on the two speaker's f0, such that when attention is focused on one speaker's f0, the other speaker's f0 would be inhibited.

## Supporting information

**S1 Fig. Two possible modes to achieve top-down lateral inhibition, resulting in sharpening of the spatial tuning of the attended channel.** A) Direct inhibition from I2 to I neuron, and B) Feed forward inhibition from I2 -> E -> I neuron. Voltage traces of each model neuron under the passive or attending condition is shown on first two columns of the grid. The final column shows spatial tuning of the attended channel in the passive vs attending conditions. (TIF)

**S2 Fig. AIM mechanism can weaken inhibitory regions of frequency-dependent PSTHs.** A) Within-channel inhibition (S neuron) is added to the AIM network for this simulation to induce suppression of activity relative to spontaneous firing of E neurons, thereby creating an inhibitory region in the frequency-dependent PSTH. Note that S neurons are distinct from I neurons, which inhibit other frequency channels. B) Frequency-dependent PSTHs. Blue regions in PSTH indicate lower firing rate relative to the spontaneous firing rate. When attention is turned on and I2 neuron is turned off, the E neuron is released from inhibition, resulting in the weakening of the inhibitory region. Simulation parameters are listed in the table below. (TIF)

**S1 Table. Model parameters for attention-induced weakening of inhibitory regions in frequency-dependent PSTHs as shown in S2 Fig.**
(XLSX)

## Acknowledgments

The authors thank Larry Abbott and John Middlebrooks for comments on the manuscript.

## Author Contributions

**Conceptualization:** Kamal Sen.

**Formal analysis:** Kenny F. Chou.

**Funding acquisition:** Kamal Sen.

**Methodology:** Kenny F. Chou.

**Software:** Kenny F. Chou.

**Supervision:** Kamal Sen.

**Validation:** Kenny F. Chou.

**Visualization:** Kenny F. Chou.

**Writing – original draft:** Kenny F. Chou, Kamal Sen.

**Writing – review & editing:** Kenny F. Chou, Kamal Sen.

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
