## [Decision Letter · Decision Letter 0]

12 May 2021

Dear Chou,

Thank you very much for submitting your manuscript "AIM: A network model of attention in auditory cortex" for consideration at PLOS Computational Biology.

As with all papers reviewed by the journal, your manuscript was reviewed by members of the editorial board and by several independent reviewers. In light of the reviews (below this email), we would consider a significantly-revised version that takes into account the reviewers' comments, and in particular the comments of reviewer 1 on the inhibitory portions of the STRFs and their modulation.

We cannot make any decision about publication until we have seen the revised manuscript and your response to the reviewers' comments. Your revised manuscript is also likely to be sent to reviewers for further evaluation.

Sincerely,

Boris S. Gutkin

Associate Editor

PLOS Computational Biology

Lyle Graham

Deputy Editor

PLOS Computational Biology

Reviewer's Responses to Questions

**Comments to the Authors:**

Reviewer #1: This paper presents a model for spectral and spatial tuning in the auditory cortex that is able to adjust this tuning based on a proposed attention mechanism. The model is inspired by known circuitry in cortex and may well be generalizable to other senses and regions of the brain. The spatial and spectral tuning essentially operate in the same way, with both represented as angle of incidence and tonotopic organizations across neurons, respectively. Temporal aspects such as modulation are not considered.

A major concern with the paper relates to attentional changes in spectral tuning. This is exemplified in Figure 4 where it is clear that the AIM model does not replicate the inhibitory regions of the STRFs seen in experimental studies. While it is claimed that the AIM model reproduces the experimental results, it is only reproducing half of the results, which significantly simplifies the problem. This hides the fact that, while the model produces an increase in response as well as a sharpening of the response for an excitatory region in the STRF, the model would also increase the response for an inhibitory region. Fritz and colleagues have shown in many publications that an excitatory region should be increased with attention, but an inhibitory region should be decreased, suppressed, or removed. Furthermore, David et al. (2012, www.pnas.org/cgi/doi/10.1073/pnas.1117717109) showed that different training paradigms can produce the opposite effect, whereby an inhibitory region is increased, but an excitatory region is decreased, suppressed, or removed. The point here is that top-down control will have opposing effects on excitatory and inhibitory regions, but the mechanism proposed in this paper will increase both types of regions. It is a concern that this has not been mentioned or discussed.

Introduction

The introduction is very sparse and does not provide much background information on the state of the art of modelling of the auditory system. While some other models are mentioned in the Discussion, there is value in highlighting the key achievements and shortcomings of existing models to inspire the development of the AIM approach.

Results

The figures are generally rather difficult to interpret especially in comparing the model results with experimental results. The tuning plots for the AIM model and experimental results should be shown with the same time scales so that they can be more directly and easily compared. A key difference in spatial tuning (Fig 2) is that the duration of response to stimuli are very different between AIM and experiment. While I don’t think this is necessarily a problem, it is important to be able to easily see this and the differences explained. Furthermore, it would be really useful to see peristimulus time histograms of the experimental data for direct comparison with the AIM output. Similarly, in Figs 4 and 5, the same suggestions apply.

Line 131: Reference to Fig 4 should be Fig 3.

Figure 4: Explain what f_T and f_B are to help the reader.

Since the model is seeking to model many spatial channels, the functional implications of spatial attention would benefit with presentation of model results for less extreme differences than 90 degrees. While 0 and 90 could be used as an example, the quantitative results can be presented for many more angles to illustrate the strengths and limitation of the model in this regard.

The result for spectral attention is limited to the spectrum of the fundamental voicing frequency, F0, with a throwaway line that this may contribute to speech segregation. Further justification of this as a sufficient, or at least major, mechanism should be discussed especially in relation to the observations experimentally that the attentional mechanisms apply across a much wider range of frequency and that F0 is also largely a temporal phenomenon (seen as striations in spectrograms). The impact on the remaining spectral information should also be quantified in terms of how much the speech at higher frequencies well above F0 is being suppressed by the model. Furthermore, similar to the comment above, quantitative reporting across many speech tokens with different F0 could be reported including speech where F0 changes substantially across the duration of the token.

Discussion

The impact of not modelling the inhibitory regions of the STRFs should be discussed.

On p. 17, a case is made that there is a distinction between the type of inhibition used in AIM vs. “classical” lateral inhibition. I suggest that discussion should be added of WHY this is important to consider.

Methods

The neural model is poorly described. There are many variables that are not included in the formulation that is presented, including i_tonic, V_thresh, V_spike, V_reset, i_e, as well as the formulation of netcon. What is the spiking-and-reset mechanism and how it is handled? Why is there an action potential voltage for a simply LIF model? The synaptic input current equation includes u(t), which is zero for t<0 and positive otherwise and so does not represent events occurring at t > 0. I can understand how it is used but the notation is imprecise. I suggest that labels use non-italics, reserving italics only for variables and functions. Table 1 include 1/10 for g_leak – it is not clear why it is given in this way.

The functional listening method does not describe how the measurements of outcomes were made.

Reviewer #2: The authors propose a network model for auditory attention, with a mechanism called attentional inhibitory modulation (AIM), in which an additional group of inhibitory neurons (called I2) are added to the conventional E-I networks and could emulate spatial and spectral auditory attention through disinhibition of the E-I network. In particular, they consider spectral selectivity, spectral sharpening, secondary spectral selectivity, and a cocktail party scenario, and replicate some of the existing experimental findings using the AIM mechanism. They further hypothesize that the top-down inputs could come from VIP neurons, I2 neurons could be SOM neurons, and I neurons could be SOM or PV neurons. Also, the different "modes" of the network are achieved by neuromodulatory mechanisms such as those mediated by acetylcholine and norepinephrine.

This is a nice theoretical contribution and the predictions of the E-I network enhanced with the AIM mechanism are impressive in replicating various existing phenomenological findings involving auditory attention. I have the following comments/suggestions for the authors:

1) The hypothesis that switching from the "passive"/"monitor" to "attentive" mode is facilitated by modulatory mechanisms (e.g., cholinergic) is relevant to long-term effects, i.e., the time constants of ACh and NE mechanisms are in the order of minutes. However, switching from passive to attentive could happen in the order of seconds or sub-second. Could the author comment on potential mechanisms that could result in rapid changes of in the top-down input (i.e., VIP neurons) as well? Could it be the case that the top-down input is coming from SOM neurons, and the VIP neurons control the top-down input?

2) Somehow related to the previous comment, given that the underlying neurons are modeled as LIF, it would be helpful to see how "turning off" an I2 neuron at a specific angle (in the spatial model) or frequency would transiently change the tuning properties. I suggest adding some snapshots of the STRFs vs. time after a particular I2 neuron is turned off, corresponding to the results of Figs. 2, 4, and 5.

3) How are the STRFs in Figs. 2, 4, and 5 estimated? Are they estimated based on multiple repeated trials and using reverse correlation on the C responses? Please explain.

4) In the cocktail party simulation (Figs. 6 and 7), the model of attending to the male vs. female F0 seems to be too simplistic, given that a growing list of studies suggest that tracking the slow spectrotemporal modulations of speech (e.g., envelope) are key in maintaining auditory attention. Could the authors use a similar model, but consider the average of spectral power across channels as the input to neurons, instead of just F0?

5) Also in the cocktail party setting, how are the I to E connection profiles chosen? Is the I to E connectivity determined based on an assumed tonotopic map/preference, or just random connections? Please clarify.

6) In lines 490-492, it is stated that the effect of ACh receptors is modeled by changing the global gain of E-C and IC-E connections. How are these gains (i.e., 0 and 2.5) determined? If these changes are not applied, or other gain values are used, would the resulting spectral tuning profiles change dramatically? It seems that the top-down signal to I2 neurons suffices for the network model to exhibit the changes in tuning. Could the authors explain why these global gain changes are necessary here?

**Have the authors made all data and (if applicable) computational code underlying the findings in their manuscript fully available?**

Reviewer #1: Yes

Reviewer #2: **No: **Codes are stated to be available upon request.

PLOS authors have the option to publish the peer review history of their article (what does this mean?). If published, this will include your full peer review and any attached files.

Reviewer #1: No

Reviewer #2: No
---

## [Decision Letter · Decision Letter 1]

30 Jul 2021

Dear Chou,

Thank you very much for submitting your manuscript "AIM: A network model of attention in auditory cortex" for consideration at PLOS Computational Biology. As with all papers reviewed by the journal, your manuscript was reviewed by members of the editorial board and by several independent reviewers. The reviewers appreciated the attention to an important topic.

Based on the reviews, we are likely to accept this manuscript for publication, providing that you modify the manuscript according to the review recommendations. Notably the authors must address fully the comments by reviewer 1.

Sincerely,

Boris S. Gutkin

Associate Editor

PLOS Computational Biology

Lyle Graham

Deputy Editor

PLOS Computational Biology

[LINK]

Reviewer's Responses to Questions

**Comments to the Authors:**

Reviewer #1: Most of my concerns have been addressed well in this revised version of the manuscript.

My major remaining concern is that the model is not able to replicate the complex STRFs as seen in experiments, which include both excitatory and inhibitory sub-regions. The additional supplementary figure shows how an inhibitory-only STRF can be modified by the top-down influences, which is good, but actually further highlights this shortcoming of the model. This is despite the Author Summary stating that the model can account for "diverse" experimental observations, which implies a much more complete model. It is not clear how the model might be expanded to be able to model experimental data with the greater level of detail.

The plots in Figure 2 are better with the same time scales. However, this further highlights the differences in temporal durations of the responses between the model and the Lee and Middlebrooks data. I asked for these duration differences to be explained but this was not addressed in the response or revised manuscript.

I like the addition of Fig. 6E. However, it is very hard to understand this figure without very careful re-reading and I still am not sure that I understand it. I advise a fuller explanation be added of the method for creating the figure and text to help the reader interpret the figure.

There are a number of typos in the paper that need to be fixed, e.g.:

line 33: in [the] midst

70: an animal [is]

76: compute salience

many semicolons are used where commas would be more appropriate

Reviewer #2: The authors have done a great job addressing my previous comments/suggestions. I have no further comments at this point.

**Have the authors made all data and (if applicable) computational code underlying the findings in their manuscript fully available?**

Reviewer #1: Yes

Reviewer #2: Yes

PLOS authors have the option to publish the peer review history of their article (what does this mean?). If published, this will include your full peer review and any attached files.

Reviewer #1: No

Reviewer #2: No

Figure Files:

Data Requirements:

Reproducibility:

References:

---

## [Editor Report · Decision Letter 2]

18 Aug 2021

Dear Chou,

We are pleased to inform you that your manuscript 'AIM: A network model of attention in auditory cortex' has been provisionally accepted for publication in PLOS Computational Biology.

Best regards,

Boris S. Gutkin

Associate Editor

PLOS Computational Biology

Lyle Graham

Deputy Editor

PLOS Computational Biology

---

## [Editor Report · Acceptance letter]

23 Aug 2021

PCOMPBIOL-D-20-02294R2 

AIM: A network model of attention in auditory cortex

Dear Dr Chou,

I am pleased to inform you that your manuscript has been formally accepted for publication in PLOS Computational Biology. Your manuscript is now with our production department and you will be notified of the publication date in due course.

With kind regards,

Andrea Szabo
